# Insecticidal Activity of *Bacillus thuringiensis* Proteins against Coleopteran Pests

**DOI:** 10.3390/toxins12070430

**Published:** 2020-06-29

**Authors:** Mikel Domínguez-Arrizabalaga, Maite Villanueva, Baltasar Escriche, Carmen Ancín-Azpilicueta, Primitivo Caballero

**Affiliations:** 1Institute for Multidisciplinary Research in Applied Biology-IMAB, Universidad Pública de Navarra, 31192 Mutilva, Navarra, Spain; mikel.dominguez@unavarra.es (M.D.-A.); maite.villanueva@unavarra.es (M.V.); 2Bioinsectis SL, Avda Pamplona 123, 31192 Mutilva, Navarra, Spain; 3Departamento de Genética/ERI BioTecMed, Universitat de València, Burjassot, 46100 València, Spain; baltasar.escriche@uv.es; 4Departamento de Ciencias, Universidad Pública de Navarra, 31006 Pamplona, Spain; ancin@unavarra.es

**Keywords:** *Bacillus thuringiensis* proteins, coleopteran pests, insecticidal activity, structure, mode of action

## Abstract

*Bacillus thuringiensis* is the most successful microbial insecticide agent and its proteins have been studied for many years due to its toxicity against insects mainly belonging to the orders Lepidoptera, Diptera and Coleoptera, which are pests of agro-forestry and medical-veterinary interest. However, studies on the interactions between this bacterium and the insect species classified in the order Coleoptera are more limited when compared to other insect orders. To date, 45 Cry proteins, 2 Cyt proteins, 11 Vip proteins, and 2 Sip proteins have been reported with activity against coleopteran species. A number of these proteins have been successfully used in some insecticidal formulations and in the construction of transgenic crops to provide protection against main beetle pests. In this review, we provide an update on the activity of Bt toxins against coleopteran insects, as well as specific information about the structure and mode of action of coleopteran Bt proteins.

## 1. Introduction

The use of entomopathogenic microorganisms as biological control agents has become one of the most effective alternatives to chemical pest control. Among all, the Gram-positive bacterium *Bacillus thuringiensis* (Bt) is the most important entomopathogenic microorganism used to date in crop protection. This bacterium is widely distributed in various ecological niches, such as water, soil, insects, and plants [1]. The feature that distinguishes *B. thuringiensis* from other members of the *Bacillus* group is the capacity to produce parasporal crystalline inclusions. These crystals are composed of proteins (Cry and Cyt) which are toxic against an increasing number of insect species from the orders Lepidoptera, Diptera, Coleoptera, Hymenoptera, and Hemiptera, among others, as well as against other organisms such as mites [2] and nematodes [3]. Bt also synthesizes insecticidal toxins associated with the vegetative growth phase, named Vip (vegetative insecticidal protein) and Sip (secreted insecticidal protein), which are secreted into the growth medium [4]. These toxins are uniquely specific, safe, and completely biodegradable, and have been used for more than 60 years as an alternative to chemical insecticides [5]. Products based on Bt isolates are the most successful microbial insecticides, with current worldwide benefits estimated at $8 billion annually [6]. However, not all Bt proteins are designated as toxins, for example, some parasporins do not have known insect targets, although they are toxic to human cancer cells [7]. The insecticidal activity of Bt toxins has also been transferred to crop plants through genetic engineering, providing very high protection levels against injurious pests and decreasing the use of chemical insecticides in many instances [8,9]. The success of these insecticidal proteins has fuelled the search for new Bt isolates and proteins that can render novel insecticidal agents with different specificities.

Since Schnepf and Whiteley cloned the first *cry* gene in the early 1980′s [10], many others have been described and are now classified according to Bt Toxin Nomenclature, that consists of four ranks based on amino acid sequence identity [11]. To date, the Bt Toxin Nomenclature Committee [12] has reported at least 78 Cry protein groups, from Cry1 to Cry 78, divided into at least three phylogenetically non-related protein subfamilies that may have different modes of action: the three-domain Cry toxins (3-domain), the mosquitocidal Cry toxins (Etx_Mtx2), Toxin_10 proteins, and alpha-helical toxins (reviewed in [13,14]).

The largest group, with more than 53 Cry toxin subgroups, is the 3-domain Cry toxin group. Even though the sequence identity among these proteins is low, the overall structure of the three domains is quite similar, providing proteins with different specificities but with quite similar modes of action [15]. Thus, proteins such as Cry1Aa (lepidopteran specific) and Cry3Aa (coleopteran specific) have a 32.5% identity but a structural similarity as high as 98% [16]. Phylogenetic analysis shows that the great variability in the insecticidal activity of this 3-domain group has resulted from the independent evolution of the three structural domains as well as from the swapping of domain III between different toxins [15].

Due to their feeding habits, many species of coleoptera cause serious damage to both cultivated plants and stored products, leading to significant economic losses in all regions of the world [17,18]. Both larvae and adults have strong jaws, which enable them to feed on a wide variety of plant substrates, such as roots, stems, leaves, grains or wood [19]. Beetles represent the order of the Insecta class that includes the largest number of species. However, the studies carried out to identify toxins of *B. thuringiensis* active against beetles are far from being equal to those carried out in the order Lepidoptera. Thus, 45 Cry proteins, 2 Cyt proteins, 11 Vip proteins, and 2 Sip proteins have been reported with activity against coleopteran insects to date, of which the toxins of the Cry3 and Cry8 families have the largest host spectrum (Figure 1). In this review, we provide an update on the activity of Bt toxins against coleopteran pests.

## 2. The Crystal Coleopteran-Active Proteins

*Bt* crystal proteins (δ-endotoxins) are produced during the stationary growth phase and have been isolated from a wide range of insect pests. These crystal inclusions are mainly formed by Cry and Cyt proteins that are toxic to a wide variety of insect species. Most of the information on the insecticidal properties has been obtained for the Cry3 family, and only a few data come from other Cry families. The Cyt proteins constitute a smaller group, mainly active against dipterans, although some Cyt proteins are toxic to coleopteran pests and increase the potential of certain Cry toxins [20].

### 2.1. Protein Structure

As mentioned above, Bt Cry proteins can be basically subdivided into three different groups according to their homology and molecular structure: the 3-domain group, Etx_Mtx2 proteins, Toxin_10 proteins, and alpha-helical toxins. The 3-domain Cry proteins constitute the largest and best-studied group, although there is increasing information on the ‘non-3-domain’ and Cyt proteins.

#### 2.1.1. The 3-Domain Group Toxins

All 3-domain Cry proteins are produced as protoxins of two main sizes, a ~130 kDa protoxin and shorter one of approximately 70 kDa [16] (Figure 2). The 130 kDa proteins share a highly conserved C terminus containing 15-17 cysteine residues, which is dispensable for toxicity but necessary for the formation of intermolecular disulphide bonds during crystal formation [15,21]. This group has been mainly studied on lepidopteran toxins such as Cry1A, but also includes some coleopteran active toxins such as Cry7A and Cry8. The structure of the small protoxins is quite similar to the N-terminal half of the large toxin group. Since these do not contain the C-terminal extension, they require, in some cases, the presence of accessory proteins for crystallization [22,23]. This second group includes Cry2A, Cry11A, and some toxins active against Coleoptera, such as Cry3A or Cry3B. Proteolytic cleavage of the N-terminal peptide and the C-terminal extension (mainly in the long Cry protoxins) yields active ~ 60 kDa protease-resistant fragments [24]. The first crystal structure solved by X-Ray crystallography was the coleopteran-specific Cry3Aa [25]. Since then, the tertiary structure of other six 3-domain Cry active proteins, including Cry1Aa, Cry2Aa, Cry3Bb, Cry4Aa, Cry4Aa, Cry4Ba and Cry8Ea, has been determined [26,27,28,29,30,31]. Among all, Cry3Aa, Cry3Bb and Cry8Ea have been defined as coleopteran-active proteins (Figure 3A,B). Using the FATCAT server [32], the structural alignment between these anti-coleopteran proteins is significantly similar, despite their low sequence identity. Pardo-López et al. [16] analyzed the structural similarity between Cry1Aa and the other 3-domain Cry proteins aforementioned, indicating the same structural likeness. The marked similarity in terms of the structure of the 3-domain Cry proteins, despite the low sequence identity and the differences in specificity, has rendered different proteins with similar modes of action.

Domain I consists of six α-helix surrounding a hydrophobic helix-α5. This domain, which shares strong similarities with the structure of the pore-forming domain of α+PFTs colicin A, might be responsible for membrane penetration and pore formation [23]. The binding domain II is constituted by three antiparallel β-sheets packing together and has an important role in receptor binding affinity. Finally, domain III is a two-twisted anti-parallel β-sheet and is also involved in receptor binding and pore formation [24,34]. Although it has been demonstrated that domains I and II have co-evolved over the years, swapping by homologous recombination of domain III has also been reported [15,35]. Local alignment of coleopteran-active Cry3, Cry7, and Cry8 showed that domain I was strongly conserved while domains II and III diversified [35]. Bt might use this mechanism to get adapted to a new insect host, which may explain the great variability in the biocidal activity of the 3-domain Cry proteins.

#### 2.1.2. Non-3-domain Cry Toxins

In addition to the 3-domain Cry proteins, some unrelated Cry proteins are also designated by the Cry nomenclature: Etx_Mtx2 proteins, Toxin_10 proteins and alpha-helical toxins [4]. The structure and function of Etx_Mtx2 proteins remains unclear, although the similarities with the *Clostridium perfringens* epsilon toxin (closely related to aerolysin) seem to indicate that they may have a β-sheet-based structure and a pore-forming activity [36]. It is important to notice that, while most of them have activity by themselves, some toxins are proposed as protein complexes to induce mortality, such as the Etx_Mtx2 protein Cry23 and the Cry37 protein [37]. The crystal structure of Cry23Aa reveals a single β-stranded domain protein, with structural similarity to several β-pore forming toxins as proaerolysins, produced by other bacterial species [38]. Cry37Aa conforms to a C2 β-sandwich fold, similar to the calcium phospholipid-binding domain observed in human cytosolic phospholipase A2 (Figure 3C) [38]. Moreover, the toxins Cry34 and Cry35 have been reported to have binary activity against coleopteran insects [39,40]. Crystal structures of Cry34Ab and Cry35Ab have been published (Figure 3D). Cry35Ab, a member of Toxin_10 proteins, shows an aerolysin-like fold, containing a β-trefoil N-terminal domain similar to the carbohydrate-binding domain in Mtx1. Cry34Ab is also a member of the aerolysin family with a β-sandwich fold, common among other cytolytic proteins [41].

#### 2.1.3. Cyt Proteins

Similar to the Cry proteins, Cyt proteins are produced as protoxins with a proteolytically activated size of around 25 kDa [20]. As with some Cry proteins, the tertiary structure of some Cyt proteins has already been solved. Cyt1Aa [42], Cyt2Aa [43] and Cyt2Ba [44] show a similar structure composed of a single α−β-domain, with two outer layers of α-helix wrapped around a β-sheet (Figure 3E). Studies performed with peptides of Cyt1A show that α-helix peptides are major structural elements involved in membrane interaction [45] and also in the oligomerization process [46], while the β-strand forms an oligomeric pore with a β-barrel structure into the membrane [43].

### 2.2. Insecticidal Activity

The vast majority of Cry proteins described to date are toxic to lepidopteran pests, but there are also a few crystal proteins toxic to either coleopteran or dipteran insects, and a small number are toxic to nematodes [47]. Currently, 45 Bt crystal proteins, including Cry, Cyt or binary proteins, have been tested against different coleopteran insects (Table 1).

#### 2.2.1. Host Range

Cry proteins are toxic to a large number of beetle pests. Mainly, the Cry3 group, the best-studied one, has been described with activity against most of the coleopteran species assayed. These Cry proteins, encoded by *cry3* genes, were first discovered in the subspecies *tenebrionis* [48] and *san diego* [49] although, years later, both strains turned out to be the same subsp. [50]. Since then, more isolates like Bt subsps. *tolworthi, kumamotoensis,* or *kurstaki* have been reported to encode a *cry3* gene [51,52]. Owing to the well-known activity in important coleopteran pests, such as *Leptinotarsa decemlineata* (Coleoptera: Chrysomelidae) or *Diabrotica* spp. (Coleoptera: Chrysomelidae), some of these isolates have been developed as bioinsecticides for beetle control [47]. Cry3Aa, Cry3Ba, Cry3Bb and Cry3Ca proteins have shown activity against most major coleopteran families, including *Chrysomelidae*, *Curculionidae*, *Scarabaeidae,* and *Tenebrionidae*, among others (Table 1). Although Cry3 proteins are the most effective Bt toxins against chrysomelid beetles, the widespread use of Cry3-based insecticides and Bt crops carries the risk of selecting insect biotypes tolerant to that proteins. The appearance of resistant populations of the chrysomelids *L. decemlineata*, *Chrysomela scripta* under laboratory conditions or *Diabrotica* spp. to Bt maize have been reported [53,54,55].

Cry7 and Cry8 groups are comparatively less active on chrysomelids, but they represent a serious alternative to Cry3 proteins. Cry7Aa, formerly known as CryIIIC, is very toxic to Cylas species (Coleoptera: Brentidae) [56], even more than Cry3 protein, but it has no negative effects against *Anthonomus grandis* (Coleoptera: Curculionidae) or *D. undecimpuntata* [52]. Moreover, toxicity to Colorado potato beetle has been reported, but only after in vitro solubilization [52], which was countered by a recent report of a Cry7Aa-type protoxin which is active against *L. decemlineata* without any previous solubilization step [57]. Solubilized Cry7Ab is active against *Henosepilachna vigintiomaculata* (Coleoptera: Coccinellidae) and *Acanthoscelides obtectus* (Coleoptera: Chrysomelidae), but not against *Anomala corpulenta* (Coleoptera: Scarabaeidae) or *Pyrrhalta aenescens* (Coleoptera: Chrysomelidae) [57,58]. Cry8-type proteins are toxic to a large number of coleopteran pests, particularly against species in the Scarabaeidae family [59,60,61]. Furthermore, Cry8A and Cry8B proteins have shown activity against the chrysomelids *L. decemlineata* and *Diabrotica* spp., Cry8Ca against the tenebrionid *Alphitobius diaperinus* (Coleoptera: Tenebrionidae) [62] and Cry8Ka against the curculionid *A. grandis* [63]. Moreover, some Cry8 proteins, such as Cry8Ea, Cry8Ga or Cry8Na, are very specific, showing different activities against very closely related host species [64,65]. Cry6Aa and Cry6Ba are active against the curculionid beetles *Hypera postica* and *Hypera brunipennis*, two of the more important pests in alfalfa [66,67], as well as *D. virgifera*, which is susceptible to the activated toxin. Cry22 proteins also have activity to a wide spectrum of coleopteran insects. In particular, Cry22A and Cry22B proteins are toxic to coleopterans of the Brentidae, Chrysomelidae and Curculionidae families [56,68,69].

Generally, Bt protein groups are particularly toxic to a certain insect order. However, some proteins may be active against different orders [70]. Mainly lepidopteran proteins Cry1Ba and Cry1Ia have shown activity against the key coleopteran pests *A. grandis*, *A. obctetus*, *C. scripta* and *L. decemlineata* [71,72,73,74,75,76]. Dual activity against Lepidoptera-Coleoptera has also been demonstrated by Cry9-type proteins. Cry9 toxins exhibit strong activity against main lepidopteran pests, but Cry9Da is also toxic against the scarab *Anomala cuprea* [77]. Other example of cross-order toxicity is depicted by the dipteran toxin Cry10Aa, which can kill the Cotton boll weevil (*A. grandis*) [78]. Additionally, Cry51Aa is toxic against *Lygus* spp. (Hemiptera) and *L. decemlineata* [79] and Cry55Aa, a typical nematicidal protein, has been reported as toxic to the chrysomelid *Phyllotreta cruciferae* [80].

Binary toxins, structurally different from classical 3-domain Cry proteins [25], used to be considered as single toxins because both proteins are required to kill their target. To date, two binary complex toxins have been proposed to have activity against beetles. The coleopteran specific Cry23Aa has been assayed together with Cry37Aa protein to kill *Popillia japonica* (Coleoptera: Sacarabaeidae) and *Tribolium castaneum* (Coleoptera: Tenebrionidae) [37]. Furthermore, this protein mixture has been found to be active against *Cylas* spp. (Coleoptera: Brentidae) and *A. obtectus* [56,75]. On the other hand, Cry34 protein is only active in association with Cry35 protein [17]. Cry34 and Cry35 are closely related and are often encoded in the same operon, with coordinated function and appearance in crystals [40,81]. The Cry34/Cry35 binary proteins are mainly active against corn rootworms and have been developed for in-plant control in Bt maize [40,82].

*B. thuringiensis* Cyt proteins have an in vitro cytolytic (hemolytic) activity, hence their name, and show predominant dipteran specificity [24]. However, some of them are also toxic to coleopteran pests, such as Cyt1Aa to *C. scripta* [72] or Cyt2Ca to the chrysomelids *L. decemlineata* and *Diabrotica* spp. [83] and the curculionid *Diaprepes abbreviates* (Coleoptera: Curculionidae) [84,85]. Besides, Cyt proteins improve the activity of Cry proteins. For instance, Cyt1Aa is able to overcome high levels of resistance to Cry3Aa by *C. Scripta*, playing an important role in resistance management [72].

#### 2.2.2. Genetically Engineered Cry Genes

Recent advances in next generation sequencing and genetic engineering technologies allow the construction of new synthetic *cry* genes that increase or amplify their toxicity. The domain regions of some lepidopteran-specific proteins have been modified in an attempt to improve their specific activity or broaden their host range [15,153]. The first coleopteran hybrid protein was made by fusing the sequences located in domain III of the *cry3A* and *cry1Aa* genes, although unfortunately, it caused the loss of activity against *L. decemlineata* [154]. Nonetheless, substituting domain III of Cry3Aa with the same domain from Cry1Ab induced activity against WCR (Western corn rootworm) larvae [155]. On a different approach, a *cry3Bb1* gene was engineered with five amino acid substitutions to produce the new Cry3Bb1.11098 protein, which increased the activity of the natural protein against WCR [156]. Similarly, a Cry3A variant (eCry3.1Ab) was designed to confer novel activity against rootworms by creating a cathepsin G protease recognition site [157]. This technology has been introduced successfully in the development of transgenic plants, mainly to overcome the appearance of resistance by WCR populations [158].

### 2.3. Mode of Action

The mode of action has been mostly studied in lepidopteran insects, although it is believed to be similar between different insect orders, with some peculiarities [8]. Briefly, it is widely accepted that the process begins once the target insect ingests the protein and reaches the insect midgut, where it is solubilized and proteolytically activated. Such an activation allows toxins to first bind to their specific receptors in the host cell membrane, then to their oligomerization and, eventually, to the formation of pores in the cell membrane (Figure 4). In this multi-step mode of action, several factors may contribute to protein specificity [159].

#### 2.3.1. Solubilization and Proteolytic Processing

Once proteins reach the host midgut, they are released from their crystal package to initiate the pathogenic process. The crystals are stabilized by disulfide bridges among the C-terminal ends of the protoxins. More recently, the occurrence of 20 kbp DNA fragments with protoxins and 100–300 pb DNA fragments with in vitro proteolytic activated toxins has been established [160]. These DNA fragments have been observed to be associated with different Bt-toxins as Cry1A, Cry2A, etc., however, they have been more extensively studied on Cry8 toxins [21]. The sequence of the DNA fragments is not specific and they are located in plasmids and chromosomes [161]. Bioinformatics modelling suggests that two protoxin regions bind to major grooves and another one, combined with phosphoric acid, binds to the minor groove [162]. The associated DNA should be eliminated by the DNAses in the insect gut for the correct protein activation. In fact, DNA-protein association impairs the specific binding [163].

It is well accepted that solubilization processes are due to the environmental conditions in the susceptible insect midgut, mainly to pH values. Of note, unlike the alkaline midgut of lepidopteran and dipteran insects, beetles have an acidic midgut, suggesting that different solubilization conditions are needed for each protein [164]. For instance, the midgut fluids of *L. decemlineata* and *D. virgifera* larvae do not seem to solubilize Cry1B and Cry7Aa1, and only after a previous in vitro solubilization, these proteins become active [52,71]. However, recent reports show that Cry7Ab2 and Cry7Aa2 proteins solubilize into midgut fluids of *H. vigintioctomaculata* and *L. decemlineata* larvae, respectively, suggesting that the lack of solubilization involves more factors than pH [57,58]. Cyt proteins dissolve readily under alkaline conditions, especially at pH 8 or higher, and they are harder to solubilize in neutral or slightly acidic pHs, which occurs in coleopteran midguts [72]. Another example of the importance of crystal solubilization was published by Galitsky et al. [28]. They related that differences in toxin solubility, oligomerization and binding for the Cry3-type toxins, in addition to differences in domain III, might explain the different specificities of Cry3A and Cry3B (e.g., WCR is susceptible to Cry3Bb1 but not to Cry3A). Solubilized proteins are proteolytically activated by gut proteases, which generate the toxic three-domain fragment of about 65 kDa [33]. In Lepidoptera and Diptera species, the main proteases present in the alkaline midgut juices are serine proteases, especially trypsin and chymotrypsin proteases [165]. However, the coleopteran species use digestive proteases belonging to cysteine and aspartic proteases and serine proteases are only present in some cases [166]. The presence of different proteases may be an important factor in toxin activation specificity, and improper processing of Bt toxins can involve the development of insect resistances. It has been reported that the combination of Cry3Aa protein and certain protease inhibitors enhances the toxicity against *Rhyzopertha dominica* (Coleoptera: Bostrichidae) larvae, evidencing that protease inhibitors may play an important role in resistant pests management [110]. Moreover, the relevance of a nicking in the N-terminal end, in the alpha 1–3 of Domain I in the activated Cry3A and Cry8Da toxins, has been shown, which rendered an 8 kDa fragment to obtain a functional 54 kDa toxin for receptor binding [167].

#### 2.3.2. Binding to the Larval Epithelium

The activated toxin is able to bind to specific receptors located in the midgut epithelial cells to form an oligomeric pre-pore structure and alterations in the midgut receptors is a critical step for insect resistance appearance [159]. It has been demonstrated that Cry3Ba protein shares a binding receptor with Cry3Aa and Cry3Ca proteins, although heterologous-competition experiments show that both proteins may have other binding sites and only share one with Cry3Ba3 [168]. It has also been shown that Cry3Bb, Cry3Ca and Cry7Aa proteins competed for the same binding sites in *C. puncticollis*, so a mutation in the midgut receptor could render all three proteins ineffective [169]. To date, several specific coleopteran binding proteins have been identified. It has been shown that an ADAM metalloprotease can be considered as a Cry3Aa receptor in *L. decemlineata*, and this binding interaction improves Cry3Aa pore-formation [170]. GPI-anchored alkaline phosphatases (ALP) are also important for the Cry3Aa binding to *Tenebrio molitor* brush border membrane vesicles (BBMV) and are highly expressed when larvae are exposed to Cry3Aa [171]. In the same way, the Cry1Ba toxin binds to ALPs from *A. grandis* midgut cells [74]. Although some putative cadherines have been previously described [172,173], Fabrick et al. [127] were the first authors reporting a cadherin protein (TmCad1), cloned from *T. molitor* larval midgut as a Cry3Aa binding receptor. Furthermore, injection of *TmCad1* dsRNA into *T. molitor* larvae conferred resistance to Cry3Aa. Another truncated cadherin protein (DvCad1-CR8–10), isolated from the WCR, binds to activated Cry3Aa, Cry3Bb [118] and also Cry8Ca [62], enhancing the activity of *L. decemlineata*, *Diabrotica* spp. and *A. diaperinus*. Finally, in *T. castaeneum* larvae, a cadherine (TcCad1) and a sodium solute symporter (TcSSS) have been identified as putative Cry3Ba functional receptors, determinant for the specific Cry protein toxicity against coleopterans [174].

Studied Cry8-binding proteins revealed a difference from those confirmed previously as receptors for Cry1A or Cry3A proteins in lepidopteran and coleopteran insect species, such as aminopeptidases, cadherins or ABCC transporters [175,176]. A Cry8-like toxin without the C-terminal end has been described, which completely shared binding sites with Cry8Ga, despite only sharing 30% of the sequence, in *Holotrichia oblita*. Cry8Da tested on *Popillia japonica* BBMV, bound specifically with a 150 kDa membrane protein which shared homology with coleopteran β-glucosidases [177]. Cry8E and Cry8-like toxins showed, in *H. parallela* and *H. oblita,* binding to several different proteins. The most relevant for both insect species and Cry8 proteins were serine proteases, sodium/potassium-transporting proteins, and a transferrin-like protein [177,178].

There is evidence that some proteins work together to cause mortality in certain coleopteran species, although the mechanism of interaction between them remains unclear. In this way, it is hypothesized that Cry37 protein may facilitate linkage of channel-forming Cry23 toxin, given their homology to other binding proteins [24]. Moreover, the fact that Cry34Ab has some activity against the Western corn rootworm (WCR) on its own [150] seems to indicate that Cry35 has the role as a receptor of Cry34, which is mainly responsible for toxicity. Cyt proteins enhancing the insecticide potency of certain Cry toxins has been also observed. The Cyt1Aa protein, from Bt sub. *israelensis,* increases the activity of Cry11Aa toxin by acting as a membrane receptor [178]. Cyt1A also helps to overcome high levels of Cry3A resistance against *C. scripta* larvae [72]. Although this mechanism of action has not yet been elucidated, Cyt1A may act as a receptor of Cry3A to enhance the binding of this protein. This synergism between Cry and Cyt toxins is an excellent strategy to decrease the appearance of resistance to Cry proteins.

#### 2.3.3. Oligomerization and Pore Formation

Although it remains unclear, some studies suggest that activated toxins need to form an oligomeric structure before insertion to the membrane as a result of binding to specific receptors [16]. In fact, Cry proteins that form oligomeric structures are related to a high pore activity [33]. Oligomerization of 3-domain Cry proteins has been described for toxins active against different insect orders, such as Cry3 proteins in coleopteran larvae. In the brush border membrane of *L. decemlineata*, Cry3A, Cry3B and Cry3C form an oligomer prior to membrane insertion, generating a pre-pore structure that can be inserted into the membrane [168]. Cry3Aa oligomeric structures have also been reported after incubation of Cry3Aa protoxin with *T. molitor* BBMV [127]. The oligomeric structure eventually leads to the lytic pore formation that disrupts the midgut insect cell by osmotic shock. However, oligomerization studies of Cry1Ab and Cry1Ia proteins incubated with lepidopteran and coleopteran BBMV, as well as culture insect cells, showed that Cry1Ia oligomerization may not be a requirement for toxicity [179]. Besides, the appearance of Cry1Ab oligomers when incubated with coleopteran BBMV could be due to an improper insertion of oligomers into the membrane or the inability to induce the post-pore events in the cells [179]. Either way, susceptible insects can withstand minor damage, but greater damage destroys the epithelium of the midgut, leading to a disruption in feeding and subsequent starvation death. Additional to the toxin action, spores may pass through the channel, to colonize and germinate in the hemolymph and contribute to insect death by septicemia [1].

## 3. The Secretable Coleopteran-Active Proteins

In addition to the δ-endotoxins produced during the stationary phase, other protein compounds have been found in the culture supernatant of certain entomopathogenic *Bacillus* isolates. These proteins, produced during the vegetative growth stage of the bacterium, were designated as vegetative insecticidal proteins (Vip) [180] and secreted insecticidal proteins (Sip) [181]. Within the Vip family, *vip1* and *vip2* genes are co-transcript in a single 4 kbp operon, which render proteins of about 100 kDa (Vip1) and 50 kDa (Vip2) [171]. The absence of toxicity of the proteins alone suggests that it is a binary toxin for some members of the coleopteran [180] and hemipteran [182] orders. In contrast, Vip3 proteins are single-chain toxins with insecticidal activity against a wide range of lepidopteran species [183]. While *B. thuringiensis* is a good source of Vip proteins, these proteins have also been found in other closely related bacteria, such as *Bacillus cereus*, *Lysinibacillus sphaericus,* or *Brevibacillus leterosporus*. Currently, two Sip proteins have been described, both active against several coleopteran pests. The fact that strains harboring *sip1Aa* and *sip1Ab* genes also contain *cry3* and *cry8* genes, respectively, suggests that Sip1 proteins may have a role in the insecticidal mechanism against coleopteran insects [184].

### 3.1. Protein Structure

Vip1 and Vip2 proteins are found in the culture supernatant before cell lysis due to specific secretion [181,185]. Both proteins have an N-terminal signal peptide for secretion, commonly cleaved after the secretion process is completed [24,181]. The Vip1/Vip2 homology with other bacterial binary toxins and the fact that these proteins are codified by two genes encoded in a single operon, suggest the presence of a typical “A+B” binary toxin [24,185]. It has been proposed that Vip1, with moderate sequence identity (30%) and structural similarity with the binding C2-II *Clostridium botulinum* toxin and the toxin “B” of *Clostridium difficile*, is the binding domain that translocates Vip2, with homology to the Rho-ADP-ribosylatin exotoxin C3 of *Clostridium* spp, to the host cell [186,187]. As occurs with other related “B” compounds, Vip1 is formed by four domains involved in docking to enzymatic components, binding to specific cell surface receptors, oligomerization, and channel formation in lipid membranes [188]. Coleopteran active Sip1Aa protein contains a predicted Gram-positive consensus secretion signal [4] and exhibits 46% similarity with Mtx3 mosquitocidal toxin of *Lysinibacillus sphaericus* [184]. This homology may indicate that Sip1Aa toxicity should be caused by pore formation.

### 3.2. Insecticidal Activity

The activity of the Bt secretable toxins against coleopterans is depicted in Table 2. Currently, four Vip protein families have been identified, but only Vip1/Vip2 showed activity against coleopteran pests [189]. Vip1/Vip2 proteins have been tested against different coleopteran families but they have shown active only against the Chrysomelidae, Curculionidae, and Scarabeidae families, being particularly toxic to corn rootworms. Single Vip1 or Vip2 showed no mortality, confirming that these proteins must act together to be toxic [185]. Vip1Aa was highly toxic against *Diabrotica* spp when combined with Vip2Aa or Vip2Ab, but Vip1Ab/Vip2Ab (co-expressed in the same operon) and Vip1Ab/Vip2Aa were not active [185]. These data show the specificity of these proteins and suggest that the absence of toxicity is due to Vip1Ab. Moreover, Vip1Ba/Vip2Ba and Vip1Bb/Vip1Ba were toxic against *Diabrotica virgifera virgifera* [190] and binary Vip1Da/Vip1Ad had activity against the curculionid *A. grandis* and the chrysomelids *Diabrotica* spp and *L. decemlineata* [191]. These are the only Vip proteins active against the Colorado potato beetle. Vip1Ad/Vip2Ag binary proteins were the first report of demonstrated toxicity against any Scarabaeoidea larvae, being active against *Holotrichia parallela*, *H. oblita* and *Anomala corpulenta* [192]. Sip1Aa and Sip1Ab proteins have specific activity against coleopteran pests. Sip1Aa caused lethal toxicity for *L. decemlineata* larvae and stunting in *D. virgifera* and *D. undecimpunctata* larvae [181]. Sip1Ab was also toxic to *Colaphellus bowringi* Baly (Coleoptera: Chrysomelidae) but it did not harm *Hloltrichia diomphalia* (Coleoptera: Scarabaeidae) larvae [184], suggesting specific chrysomelid activity, although further studies are needed to determine its host range.

### 3.3. Mode of Action

The mode of action of coleopteran-specific Bt secretable proteins is poorly understood, but some information is available for this binary mechanism of action. The proposed multistep process begins with the ingestion of the two toxins by the susceptible larvae. Though the two encoded proteins are synthesized together, they are thought not to get associated in solution and reach the insect midgut as single proteins [188]. Then, the proteolytic processing by the trypsin-like proteases of the insect midgut juice of Vip1 allows the cell-bound “B” to bind to a specific membrane receptor, followed by the formation of oligomers containing seven Vip1 molecules [193]. It is at this stage when the docking between Vip1 and Vip2 translocates the toxic component (Vip2) into the cytoplasm though the “B” (Vip1) channel [188]. Recent studies in BBMVs of *H. parallela* evidenced that although Vip2Ag showed a low degree of binding on its own, the degree of binding increased when Vip1Ad was added, showing that Vip1Ad acted as a receptor to help Vip2 bind to BBMVs [194]. Once inside the cytosol, Vip2 destroys filamentous actin by blocking its polymerization and leading to cell death [195].

Sip1 proteins have no homology with Vip proteins, but Sip1A exhibits limited sequence similarity with the 36-kDa mosquitocidal Mtx3 protein of *B. sphaericus*, suggesting that toxicity is related with pore formation [181].

## 4. Bt Based Insecticides

In 1938, the first insecticide based on *B. thuringiensis* was produced and marketed under the name *Sporéine* for the control of lepidopteran insect pests [47]. Since then, sporulated cultures of *B. thuringiensis* have been used widely as foliar sprays to protect crops from insect damage. Since *B. thuringiensis* subsp. *tenebrionis* was discovered [48], it was rapidly formulated as a bioinsecticide and commercialized against the Colorado potato beetle. Bt-based insecticides to control coleopteran pests are mainly developed against chrysomelid beetles [198]. Novodor^®^ (Kenogard) uses the NB-176 strain of Bt subsp. *tenebrionis* as the active ingredient and is widely used for the control of *L. decemlineata*. However, the toxicity of this commercial product has been verified for other species of beetles, such as the chrysomelids *Chrysophtharta bimaculata, C. agricola* and *C. scripta* [199,200] under laboratory conditions. Furthermore, this product has been shown to be effective against *C. scripta* in field conditions [200], while the use of Novodor did not exert good control of the populations of *Lissorhoptrus oryzophilus* (Coleoptera: Curculionidae) [201].

To date, most of the Bt-based bioinsecticide products effectively use natural Bt strains for the control of foliar-feeding pests. However, several factors have limited their use. Usually, Bt strains have a narrow insecticidal spectrum compared with other insecticides, even when insects are closely related [202]. Advances in genetic manipulation technologies offer improvements in the efficiency of Bt-based formulates and reductions in their production costs. The development of new strains by conjugation or transduction has been used to confer natural strains with new insecticidal properties [203]. The natural Bt subsp. *kurstaki*, for example, has been modified to express several *cry3* genes and extend its host range to both lepidopteran and coleopteran pests [202]. The active ingredient in Foil^®^ is the Bt strain EG2424, expressing both Cry1Ac and Cry3A proteins, the latter of which was transferred from a Cry3Aa-encoding plasmid belonging to the Bt subsp. *morrisoni* [204]. Similarly, the Cry3-overproducing strain, EG7673, was obtained by transforming a natural strain with a recombinant plasmid containing a *cry3Bb1* gene. A formulation with this strain as the active ingredient was commercialized as Raven^®^ and was four-fold more active than the parental strain [205].

## 5. Bt-Crops

By expressing one or more Bt toxic genes in a target plant tissue transgenic insect-resistant crops, Bt crops, can be produced. Such cultivars need no further pest control measures. To date, the Bt crops extension has increased worldwide, particularly that of Bt cotton, Bt rice and Bt corn [9]. Bt plants have been created for the control of several insect pests, among others, Colorado potato beetle (*L. decemlineata*) and corn rootworms (*Diabrotica* spp.). The first human-modified pesticide-producing crop was potato, which expressed the *cry3A* gene from *B. thuringiensis* subsp. *tenebrionis* in their leaves [206]. The transgenic gene expression confers potato plants protection against the Colorado potato beetle and allows reducing insecticide applications [207]. A few years later, this Bt crop was complemented with another gene expression cassette that also provided protection against the Potato leafroll virus [208]. However, genetically modified potatoes were commercialized from 1995 to 2001, and eventually removed from the marketplace due to social concern for genetically modified crops [209].

A coleopteran-active Bt maize was designed for the control of corn rootworms, expressing a variant of the wild-type *cry3Bb1* gene from Bt subsp. *kumamotoensis* in the root tissue [210]. Currently, Bt maize hybrids express four different crystal proteins (Cry3Bb, mCry3A, Cry34Ab/35Ab and eCry3.1Ab), individually or co-expressing two toxins [211,212]. Vip1 and Vip2 proteins were also candidates to be expressed in maize plants, mainly due to the great toxicity against rootworms. However, the cytotoxic activity of the Vip2 protein has prevented the development of a Bt plant expressing this binary toxins [189]. The opportunity of expressing the toxin in a specific tissue allows minimization of the exposure of non-target fauna while increasing the control of tunneling and root pests, which are otherwise difficult to manage. However, Western corn rootworm has developed field resistance to all four currently available Bt toxins [212,213,214] as did *D. virgifera* in 2009 against Bt corn [55]. These facts show that although Bt crops have the potential to increase productivity while conserving biodiversity, resistance management programs and a better use of integrated pest management are necessary to delay resistance development as much as possible [215].

## 6. Resistance and Cross-Resistance

The widespread use of *B. thuringiensis* biopesticides, as well as the planting of millions of hectares of Bt plants to protect crops from pests, carry the risk of selecting insect biotypes that are tolerant or resistance to Bt toxins. The appearance of resistance may be due to alterations in any step involved their mode of action, from the solubilization and activation steps to the capacity of pore formation [159]. It is established that the lack of solubilization is favored by the physicochemical conditions of the midgut fluids, particularly the pH. The acidic midgut of the coleopteran insects seemed to be a limiting factor in the solubilization of Cry proteins, such as Cry1B and Cry7Aa [52,71], although recent reports seem to indicate that more factors are involved as Cry7Aa proteins are dissolved in *L. decemlineata* and *H. vigintioctomaculata* midgut fluids [57,58]. Once the Cry toxin is solubilized in the midgut, protoxins are proteolytically cleaved to activated toxins. This toxin processing depends on the presence of the right digestive enzymes in the host midgut fluid. As an example, it was observed in *D. virgifera* larvae that the Cry3Aa protein was poorly processed by its own proteases, which leads to low activity of Cry3Aa against rootworms [157]. Introduction of a chymotrypsin/cathepsin recognition site in domain I of Cry3A has been shown to enhance the bioactivity of this toxin against the western corn rootworm larvae [157].

Molecularly, the insect resistance basis is a modification or loss of the specific midgut cell membrane receptors or some mediator, which eliminates or reduces the capacity of the toxin to initiate a lethal pathway [216]. Cross-resistance between Cry toxins is often associated with sequence similarities in domains II and III, related to specific protein binding [217]. Under laboratory conditions, populations of *L. decemlineata* and *C. scripta* resistant to Cry3Aa have been described [53,54]. To date, the appearance of field resistance is still relatively low despite the extensive use of products based on the same protein, which increases the probability of resistance development.

Conversely, rootworm populations have developed resistance to all proteins used in transgenic corn. The intense selection pressure posed by the continuous exposure of insects to Bt toxins has increased the emergence of pest resistance. Since the first case of resistance to Cry3Bb1 Bt-maize in 2009, *Diabrotica* has developed resistance to Cry3Aa and Cry34/35Ab binary protein [211]. New strategies are being carried out to try to delay resistance, including a combined use of several proteins in the same Bt plant [218]. Pyramiding of two Bt proteins can delay resistance to those proteins because when insects become tolerant to one toxin, most will still be susceptible to the other toxin [211]. However, there is already evidence of cross-resistance to Cry3 proteins and even to Cry34/35, which may invalidate, in the long run, the use of all these proteins [212].

## Figures and Tables

**Figure 1 toxins-12-00430-f001:**
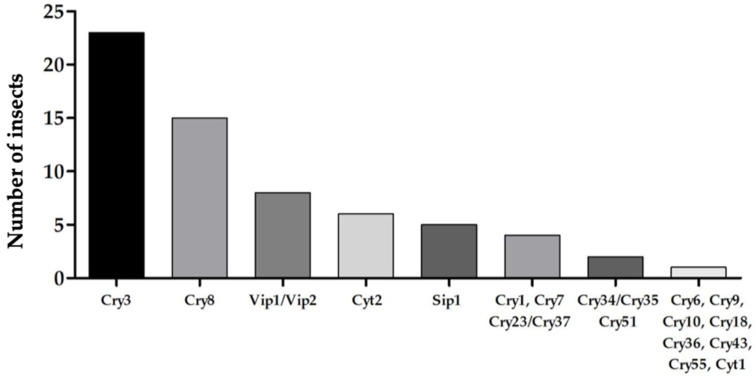
Number of susceptible coleopteran insects to Bt (*Bacillus thuringiensis*) proteins, grouped into protein families.

**Figure 2 toxins-12-00430-f002:**
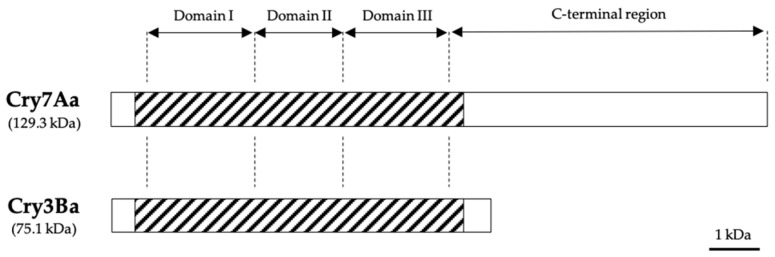
Relative length of 3-domain Cry proteins of *B. thuringiensis*, representing both main sizes of approximately 130 and 70 kDa. Dashed parts represent the activated toxin, while the white boxes represent the amino- and carboxy-terminal parts. Adapted from Bravo et al., 2007 [33].

**Figure 3 toxins-12-00430-f003:**
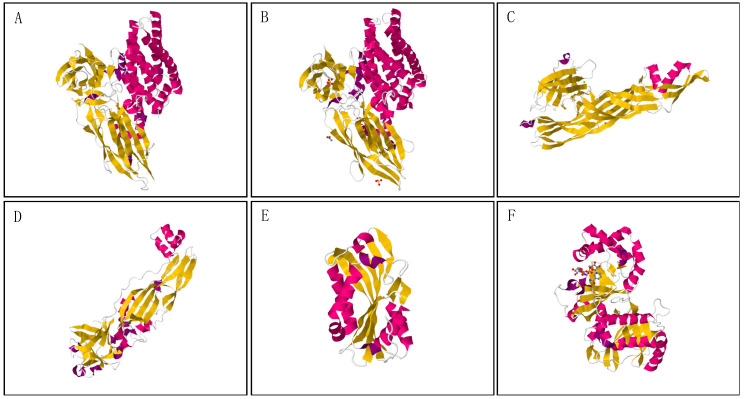
*Bacillus thuringiensis* proteins, with particular activity against coleopteran pests, for which three-dimensional structure has been predicted. (**A**) Cry3Aa (PBD accession number 4QX1); (**B**) Cry8Ea (PBD accession number 3EB7); (**C**) Protein complex Cry23Aa/Cry37Aa (PBD accession number 4RHZ); (**D**) Binary proteins Cry34Ab and Cry35Ab (PBD accession number 4JOX and 4JPO); (**E**) Cyt1Aa (PBD accession number 3RON); (**F**) Secretable protein Vip2Aa with a NAD complex (PBD accession number 1QS2).

**Figure 4 toxins-12-00430-f004:**
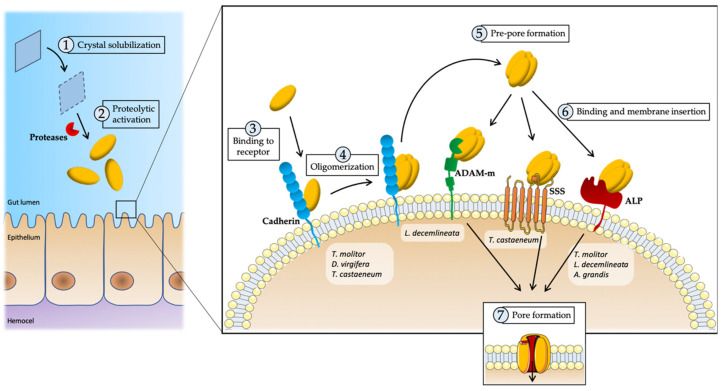
Schematic representation of the particularities in the mechanism of action of crystal proteins against coleopteran pests. (**1**) Crystal solubilizes in the acidic conditions of the coleopteran midgut lumen and (**2**) activates into toxin by proteolytic processing of the protoxin by the specific digestive enzymes, specially cysteine and aspartic proteases. (**3**) Toxins are able to bind to a first receptor (CADR), (**4**) oligomerizate and (**5**) form an oligomeric pre-pore structure that (**6**) is able to bind to a second specific receptor (ADAM metalloproteases/GPI-anchored alkaline phosphatases/sodium solute symporters). This event induces the insertion into the membrane, leading to (**7**) pore formation and finally to cell lysis.

**Table 1 toxins-12-00430-t001:** Insecticidal activity of Cry and Cyt proteins against coleopteran pests.

Crystal Type Toxin	Target Insect		Activity ^(a)^	LC_50_ ^(b)^	Reference
	Scientific Name	Family			
**Cry1Aa**	*Anoplophora glabripennis*	Cerambycidae	N		[86]
	*Apriona germari*	Cerambycidae	N		[87]
	*Epilachna varivestis*	Coccinellidae	A		[88]
	*Tribolium castaneum*	Tenebrionidae	LA		[89]
**Cry1Ab**	*Diabrotica undecimpuntata*	Chrysomelidae	N		[90]
	*Leptinotarsa decemlineata*	Chrysomelidae	N		[90]
	*Phyllotreta armoraciae*	Chrysomelidae	N		[90]
	*Adalia bipunctata*	Coccinellidae	N		[91]
	*Atheta coriaria*	Coccinellidae	N		[91]
	*Cryptolaemus montrouzieri*	Coccinellidae	N		[91]
	*Harmonia axyridis*	Coccinellidae	N		[92]
	*Anthonomus grandis*	Curculionidae	N		[90]
	*Hypera postica*	Curculionidae	N		[90]
	*Popillia japonica*	Scarabaeidae	N		[90]
**Cry1A** **c**	*Diabrotica undecimpuntata*	Chrysomelidae	N		[90]
	*Leptinotarsa decemlineata*	Chrysomelidae	N		[90]
	*Phyllotreta armoraciae*	Chrysomelidae	N		[90]
	*Hippodamia convergens*	Coccinellidae	N		[93]
	*Anthonomus grandis*	Curculionidae	N		[90]
	*Hypera postica*	Curculionidae	N		[90,94]
	*Haptoncus luteolus*	Nitidulidae	N		[95]
	*Tribolium castaneum*	Tenebrionidae	N		[96]
	*Popillia japonica*	Scarabaeidae	N		[90]
**Cry1Ah**	*Propylea japónica*	Coccinellidae	N		[97]
**Cry1Aj**	*Harmonia axyridis*	Coccinellidae	N		[92]
**Cry1Ba**	*Anoplophora glabripennis*	Cerambycidae	N		[86]
	*Acanthoscelides obtectus*	Chrysomelidae	A		[75]
	*Chrysomela scripta F*	Chrysomelidae	A	1.8 // 5.9	[71,72]
	*Leptinotarsa decemlineata*	Chrysomelidae	A	1050 // 142	[71,73]
	*Phaedon cochleariae*	Chrysomelidae	N		[98]
	*Anthonomus grandis*	Curculionidae	A	305.32	[74]
	*Asymmathetes vulcanorum*	Curculionidae	N		[99]
	*Hypothenemus hampei*	Curculionidae	A		[100]
	*Tribolium castaneum*	Tenebrionidae	N		[89]
**Cry1Ca**	*Tribolium castaneum*	Tenebrionidae	N		[89]
**Cry1Da**	*Tribolium castaneum*	Tenebrionidae	N		[89]
**Cry1Ea**	*Tribolium castaneum*	Tenebrionidae	N		[89]
**Cry1Fa**	*Cryptolestes pusillus*	Laemophloeidae	N		[17]
	*Tribolium castaneum*	Tenebrionidae	N		[17]
**Cry1Fb**	*Tribolium castaneum*	Tenebrionidae	N		[89]
**Cry1Ia**	*Acanthoscelides obtectus*	Chrysomelidae	A		[75]
	*Agelastica coerulea*	Chrysomelidae	N		[101,102]
	*Diabrotica undecimpuntata*	Chrysomelidae	N		[103]
	*Leptinotarsa decemlineata*	Chrysomelidae	A	33.7 // 10	[73,104]
	*Phaedom brassicae*	Chrysomelidae	N		[101]
	*Anthonomus grandis*	Curculionidae	A	21.5 // 230	[76,105]
	*Asymmathetes vulcanorum*	Curculionidae	N		[99]
	*Tenebrio molitor*	Tenebrionidae	N		[106]
	*Tribolium castaneum*	Tenebrionidae	N		[89]
**Cry1Ib**	*Phaedom brassicae*	Chrysomelidae	N		[101]
	*Agelastica coerulea*	Chrysomelidae	N		[101]
**Cry1Id**	*Agelastica coerulea*	Chrysomelidae	N		[102]
**Cry1Ie**	*Ceratoma trifurcata*	Chrysomelidae	N		[107]
	*Pyrrhalta aenescens*	Chrysomelidae	N		[108]
**Cry1Jb**	*Diabrotica undecimpuntata*	Chrysomelidae	N		[109]
	*Leptinotarsa decemlineata*	Chrysomelidae	N		[109]
**Cry2Aa**	*Diabrotica undecimpuntata*	Chrysomelidae	N		[93]
	*Diabrotica virgifera*	Chrysomelidae	N		[93]
	*Leptinotarsa decemlineata*	Chrysomelidae	N		[93]
	*Hippodamia convergens*	Coccinellidae	N		[93]
	*Anthonomus grandis*	Curculionidae	N		[93]
**Cry2Ab**	*Propylea japonica*	Coccinellidae	N		[97]
	*Haptoncus luteolus*	Nitidulidae	N		[95]
**Cry3Aa**	*Rhyzophertha dominica*	Bostrichidae	A	1.17 μg/mg	[110]
	*Cylas brunneus*	Brentidae	A	1.88 μg/g	[56]
	*Cylas puncticollis*	Brentidae	A	1.99 μg/g	[56]
	*Apriona germari*	Cerambycidae	A		[94,111]
	*Acanthoscelides obtectus*	Chrysomelidae	A		[75]
	*Agelastica alni*	Chrysomelidae	A		[112]
	*Brontispa longissimi*	Chrysomelidae	A	0.475 mg/mL	[113]
	*Chrysomela tremulae*	Chrysomelidae	A		[114]
	*Chrysomela scripta F*	Chrysomelidae	A		[115]
	*Chrysomela scripta F*	Chrysomelidae	A	2.22 // 1.8	[71,72]
	*Colaphellus bowringi*	Chrysomelidae	A	2.68 // 1.33	[116,117]
	*Crioceris quaturdicerumpunctata*	Chrysomelidae	A	3.82	[117]
	*Diabrotica undecimpuntata*	Chrysomelidae	N		[90,118]
	*Diabrotica virgifera*	Chrysomelidae	N		[118,119]
	*Leptinotarsa decemlineata*	Chrysomelidae	A	1.84 // 3.56	[73,118]
	*Phaedom brassicae*	Chrysomelidae	A	1.11	[117]
	*Phaedon cochleariae*	Chrysomelidae	A		[120]
	*Phyllotreta armoraciae*	Chrysomelidae	N		[90]
	*Plagiodera versicolora*	Chrysomelidae	A	1.13 // 3.09	[18]
	*Pyrrhalta aenescens*	Chrysomelidae	A	0.22 mg/ml	[121]
	*Pyrrhalta luteola*	Chrysomelidae	A	0.12 μg/cm^2^	[49]
	*Adalia bipunctata*	Coccinellidae	N		[91]
	*Atheta coriaria*	Coccinellidae	N		[91]
	*Cryptolaemus montrouzieri*	Coccinellidae	N		[91]
	*Epilachna varivestis*	Coccinellidae	A		[88]
	*Anthonomus grandis*	Curculionidae	N		[90]
	*Asymmathetes vulcanorum*	Curculionidae	N		[99]
	*Hypera postica*	Curculionidae	N		[90]
	*Hypothenemus hampei*	Curculionidae	A		[100]
	*Myllocerus undecimpustulatus*	Curculionidae	A	152 ng/cm^2^	[122]
	*Premnotrypes vorax*	Curculionidae	LA		[123]
	*Sitophilus oryzae*	Curculionidae	A		[124]
	*Amphimallon solstitiale*	Scarabaeidae	A		[112]
	*Anomala corpulenta*	Scarabaeidae	N		[116]
	*Melontha melontha*	Scarabaeidae	A		[112]
	*Popillia japonica*	Scarabaeidae	N		[90]
	*Alphitobius diaperinus*	Tenebrionidae	A	9.58 // 8 μg/cm^2^	[62,125]
	*Tribolium castaneum*	Tenebrionidae	N		[96,110]
	*Tribolium castaneum*	Tenebrionidae	A	0.46 g/10 g	[89]
	*Tenebrio molitor*	Tenebrionidae	A	11.4 μg/larve	[126]
	*Tenebrio molitor*	Tenebrionidae	A		[110,127]
**Cry3Ba**	*Cylas brunneus*	Brentidae	A	1.304 μg/g	[56]
	*Cylas puncticollis*	Brentidae	A	1.273 μg/g	[56]
	*Chrysomela scripta F*	Chrysomelidae	A		[115]
	*Diabrotica undecimpuntata*	Chrysomelidae	A	107 ng/mm^2^	[128]
	*Leptinotarsa decemlineata*	Chrysomelidae	A	1.35 ng/mm^2^	[128]
	*Epilachna varivestis*	Coccinellidae	N		[88]
	*Popillia japonica*	Scarabaeidae	A	1	[37]
	*Tribolium castaneum*	Tenebrionidae	A	1.60 g/10 g	[89]
	*Tribolium castaneum*	Tenebrionidae	A	13.55 mg/mL	[37,96]
**Cry3Bb**	*Cylas brunneus*	Brentidae	A	1.83 μg/g	[56]
	*Cylas puncticollis*	Brentidae	A	1.82 μg/g	[56]
	*Anoplophora glabripennis*	Cerambycidae	N		[86]
	*Diabrotica undecimpuntata*	Chrysomelidae	A	9.49 // 1.18	[118,129]
	*Diabrotica virgifera*	Chrysomelidae	A	2.10 // 5.18	[118,129]
	*Leptinotarsa decemlineata*	Chrysomelidae	A	6.86 // 6.54	[118,129]
	*Alphitobius diaperinus*	Tenebrionidae	A	26.52 // 50 μg/cm^2^	[62,125]
**Cry3Ca**	*Cylas brunneus*	Brentidae	A	0.69 μg/g	[56]
	*Cylas puncticollis*	Brentidae	A	0.57 μg/g	[56]
	*Leptinotarsa decemlineata*	Chrysomelidae	A	0.7 // 320.13	[130,131]
	*Tribolium castaneum*	Tenebrionidae	N		[96]
**Cry6Aa**	*Diabrotica virgifera*	Chrysomelidae	A	77 µg/cm^2^	[66,119]
	*Hypera brunneipennis*	Curculionidae	A		[66]
	*Hypera postica*	Curculionidae	A		[66]
**Cry6Ba**	*Hypera postica*	Curculionidae	A	280 ng/μl	[94]
**Cry7Aa**	*Cylas brunneus*	Brentidae	A	0.44 μg/g	[56]
	*Cylas puncticollis*	Brentidae	A	0.34 μg/g	[56]
	*Anoplophora glabripennis*	Cerambycidae	N		[86]
	*Diabrotica undecimpuntata*	Chrysomelidae	N		[52]
	*Leptinotarsa decemlineata*	Chrysomelidae	A	13.1 // 18.8	[52,57]
	*Anthonomus grandis*	Curculionidae	N		[52]
**Cry7Ab**	*Acanthoscelides obtectus*	Chrysomelidae	A		[75]
	*Ceratoma trifurcata*	Chrysomelidae	N		[107]
	*Colaphellus bowringi*	Chrysomelidae	A	293.79	[132]
	*Pyrrhalta aenescens*	Chrysomelidae	N		[58]
	*Henosepilachna vigintioctomaculata*	Coccinellidae	A	209	[58,133]
	*Anomala corpulenta*	Scarabaeidae	N		[58]
	*Tribolium castaneum*	Tenebrionidae	LA		[89]
**Cry8Aa**	*Leptinotarsa decemlineata*	Chrysomelidae	A		[134]
	*Cotinis* spp	Scarabaeidae	A		[135]
	*Tribolium castaneum*	Tenebrionidae	LA		[89]
**Cry8Ab**	*Holotrichia oblita*	Scarabaeidae	A	5.72 μg/g	[136]
	*Holotrichia parallela*	Scarabaeidae	A	2.00 μg/g	[136]
	*Tenebrio molitor*	Tenebrionidae	N		[136]
**Cry8Ba**	*Diabrotica virgifera*	Chrysomelidae	A		[119]
	*Cotinis* spp	Scarabaeidae	A		[137]
	*Cyclocephala borealis*	Scarabaeidae	A		[135]
	*Cyclocephala pasadenae*	Scarabaeidae	A		[135]
	*Popillia japonica*	Scarabaeidae	A		[135]
**Cry8Bb**	*Diabrotica undecimpuntata*	Chrysomelidae	A		[138]
	*Diabrotica virgifera*	Chrysomelidae	A		[138]
	*Leptinotarsa decemlineata*	Chrysomelidae	A		[138]
**Cry8Ca**	*Anoplophora glabripennis*	Cerambycidae	N		[86]
	*Colaphellus bowringi*	Chrysomelidae	N		[116]
	*Leptinotarsa decemlineata*	Chrysomelidae	N		[116]
	*Epilachna varivestis*	Coccinellidae	A		[88]
	*Anomala corpulenta*	Scarabaeidae	A	1.75 × 10 × 10^8^ CFU/g	[116]
	*Anomala corpulenta*	Scarabaeidae	A	1.6 × 10 × 10^8^ CFU/g	[139,140]
	*Anomala cuprea*	Scarabaeidae	A		[141]
	*Anomala exoleta*	Scarabaeidae	A		[142]
	*Holotrichia parallela*	Scarabaeidae	A	9.24 × 10 × 10^8^ CFU/g	[140]
	*Popillia japonica*	Scarabaeidae	A	12.3 μg/g	[35]
	*Alphitobius diaperinus*	Tenebrionidae	A	7.71 // 10 μg/cm^2^	[62,125]
	*Tribolium castaneum*	Tenebrionidae	N		[89]
**Cry8Da**	*Anomala cuprea*	Scarabaeidae	A		[143]
	*Anomala orientalis*	Scarabaeidae	A		[143]
	*Popillia japonica*	Scarabaeidae	A	17.0 μg/g	[35,143]
**Cry8Db**	*Popillia japonica*	Scarabaeidae	A	19.6 μg/g	[35]
**Cry8Ea**	*Plagiodera versicolora*	Chrysomelidae	A		[144]
	*Anomala corpulenta*	Scarabaeidae	A		[140]
	*Holotrichia parallela*	Scarabaeidae	A	0.9 × 10 × 10^8^ CFU/mL	[59,64,144]
	*Popillia japonica*	Scarabaeidae	A		[144]
	*Tenebrio molitor*	Tenebrionidae	N		[64]
	*Tribolium castaneum*	Tenebrionidae	N		[64,89]
**Cry8Fa**	*Anomala corpulenta*	Scarabaeidae	N		[59]
	*Holotrichia oblita*	Scarabaeidae	N		[59]
	*Holotrichia parallela*	Scarabaeidae	N		[59]
	*Tribolium castaneum*	Tenebrionidae	N		[89]
**Cry8Ga**	*Holotrichia oblita*	Scarabaeidae	N		[60]
	*Holotrichia parallela*	Scarabaeidae	N		[60]
**Cry8Ka**	*Anthonomus grandis*	Curculionidae	A	2.83–8.93	[63]
**Cry8Na**	*Anomala corpulenta*	Scarabaeidae	N		[65]
	*Holotrichia oblita*	Scarabaeidae	N		[65]
	*Holotrichia parallela*	Scarabaeidae	A	3.18 × 10 × 10^10^ CFU/g	[65]
**Cry8Sa**	*Holotrichia serrata (F.)*	Scarabaeidae	A		[145]
**Cry9Bb**	*Diabrotica undecimpuntata*	Chrysomelidae	N		[146]
	*Diabrotica virgifera*	Chrysomelidae	N		[146]
	*Leptinotarsa decemlineata*	Chrysomelidae	N		[146]
	*Anthonomus grandis*	Curculionidae	N		[146]
**Cry9Da**	*Anomala cuprea*	Scarabaeidae	A		[77]
	*Tribolium castaneum*	Tenebrionidae	N		[89]
**Cry10Aa**	*Anthonomus grandis*	Curculionidae	A	7.12	[78]
**Cry14Aa**	*Tribolium castaneum*	Tenebrionidae	LA		[89]
**Cry15Aa**	*Leptinotarsa decemlineata*	Chrysomelidae	N		[147]
**Cry18Aa1**	*Melontha melontha*	Scarabaeidae	A		[148]
**Cry22Aa**	*Anthonomus grandis*	Curculionidae	A	0.75 μg/well	[68]
	*Tribolium castaneum*	Tenebrionidae	A	1.25 g/10 g	[89]
**Cry22Ab**	*Cylas brunneus*	Brentidae	A	1.01 μg/g	[56]
	*Cylas puncticollis*	Brentidae	A	0.78 μg/g	[56]
	*Diabrotica virgifera*	Chrysomelidae	A	39.4 μg/cm^2^	[69]
	*Diabrotica undecimpuntata*	Chrysomelidae	N		[69]
	*Leptinotarsa decemlineata*	Chrysomelidae	N		[69]
	*Anthonomus grandis*	Curculionidae	A	3.12 μg/well	[68]
**Cry22Ba**	*Diabrotica virgifera*	Chrysomelidae	N		[68]
	*Anthonomus grandis*	Curculionidae	A		[68]
**Cry23Aa/37Aa**	*Cylas brunneus*	Brentidae	A	0.46 μg/g	[56]
	*Cylas puncticollis*	Brentidae	A	0.42 μg/g	[56]
	*Acanthoscelides obtectus*	Chrysomelidae	A		[75]
	*Anthonomus grandis*	Curculionidae	A		[149]
	*Popillia japonica*	Scarabaeidae	A		[37]
	*Tribolium castaneum*	Tenebrionidae	A	6.30 μg SC/μL	[37,61]
**Cry34Aa**	*Diabrotica virgifera*	Chrysomelidae	N		[40]
**Cry34Ab**	*Diabrotica undecimpuntata*	Chrysomelidae	LA		[150]
	*Diabrotica virgifera*	Chrysomelidae	N		[40,82]
**Cry34Ac**	*Diabrotica virgifera*	Chrysomelidae	N		[40]
**Cry34Aa/35Aa**	*Diabrotica undecimpuntata*	Chrysomelidae	A	34.1 μg/well	[151]
	*Diabrotica virgifera*	Chrysomelidae	A	34 μg/cm^2^	[81,151]
**Cry34Ab/35Ab**	*Rhyzophertha dominica*	Bostrichidae	N		[17]
	*Diabrotica undecimpuntata*	Chrysomelidae	A		[150]
	*Diabrotica virgifera*	Chrysomelidae	A	3 μg/cm^2^	[40,81]
	*Oryzaephilus surinamensis*	Cucujidae	LA		[17]
	*Sitophilus oryzae*	Curculionidae	LA		[17]
	*Trogoderma variabile*	Dermestidae	N		[17]
	*Tenebrio molitor*	Tenebrionidae	LA		[17]
	*Tribolium castaneum*	Tenebrionidae	LA		[17]
	*Tribolium castaneum*	Tenebrionidae	N		[96]
**Cry34Ac/35Ac**	*Diabrotica virgifera*	Chrysomelidae	A	7 μg/cm^2^	[40,81]
**Cry34Ba/35Ba**	*Diabrotica virgifera*	Chrysomelidae	A		[39]
**Cry35Aa**	*Diabrotica virgifera*	Chrysomelidae	N		[40]
**Cry35Ab**	*Diabrotica virgifera*	Chrysomelidae	N		[40,82]
**Cry35Ac**	*Diabrotica virgifera*	Chrysomelidae	N		[40]
**Cry36A**	*Diabrotica virgifera*	Chrysomelidae	A	147.3 μg/well	[151]
**Cry37Aa**	*Tribolium castaneum*	Tenebrionidae	A	1.25 g/10 g	[89]
**Cry38Aa**	*Diabrotica virgifera*	Chrysomelidae	N		[39]
**Cry43Aa**	*Anomala cuprea*	Scarabaeidae	A		[152]
**Cry43Ba**	*Anomala cuprea*	Scarabaeidae	N		[152]
**Cry51Aa**	*Diabrotica undecimpuntata*	Chrysomelidae	N		[79]
	*Diabrotica virgifera*	Chrysomelidae	N		[79]
	*Leptinotarsa decemlineata*	Chrysomelidae	A		[79]
	*Tribolium castaneum*	Tenebrionidae	A	1.45 g/10 g	[89]
**Cry55Aa**	*Phyllotreta cruciferae*	Chrysomelidae	A		[80]
	*Tribolium castaneum*	Tenebrionidae	N		[89]
**Cyt1Aa**	*Chrysomela scripta F*	Chrysomelidae	A	132.6	[72]
**Cyt2Ca**	*Diabrotica undecimpuntata*	Chrysomelidae	A	25 μg/well	[83]
	*Diabrotica virgifera*	Chrysomelidae	A	10.8 μg/well	[83]
	*Leptinotarsa decemlineata*	Chrysomelidae	A		[83]
	*Diapepes abbreviatus*	Curculionidae	A	50.7	[84,85]
	*Popillia japonica*	Scarabaeidae	A		[83]
	*Tribolium castaneum*	Tenebrionidae	A		[83]

^(a)^ The parameter is mortality. A = active; N = not active; LA = low activity, with significant inhibition of growth; ^(b)^ LC_50_ = lethal concentration that causes 50% mortality of the insects. Data are expressed in μg/mL, unless otherwise stated. “//” separate two different values of the LC_50_.

**Table 2 toxins-12-00430-t002:** Insecticidal activity of Vip and Sip proteins against coleopteran pests.

Crystal Type Toxin	Target Insect		Activity ^(a)^	LC_50_ ^(b)^	Reference
	Scientific Name	Family			
**Sip1Aa**	*Diabrotica undecimpuntata*	Chrysomelidae	A		[181]
	*Diabrotica virgifera*	Chrysomelidae	A		[181]
	*Colaphellus bowringi*	Chrysomelidae	A	1.07	[184]
	*Leptinotarsa decemlineata*	Chrysomelidae	A	24	[181]
**Sip1Ab**	*Colaphellus bowringi*	Chrysomelidae	A	1.05	[184]
	*Hloltrichia diomphalia*	Scarabaeidae	N		[184]
**Vip1Aa**	*Diabrotica virgifera*	Chrysomelidae	N		[185]
**Vip1Ac**	*Holotrichia oblita*	Scarabaeidae	N		[196]
	*Tenebrio molitor*	Tenebrionidae	N		[195]
**Vip1ad**	*Anomala corpulenta*	Scarabaeidae	N		[192]
	*Holotrichia oblita*	Scarabaeidae	N		[192]
	*Holotrichia parallela*	Scarabaeidae	N		[192]
**Vip1Da**	*Diabrotica virgifera*	Chrysomelidae	N		[191]
**Vip2Aa**	*Diabrotica virgifera*	Chrysomelidae	N		[185]
**Vip2Ac**	*Tenebrio molitor*	Tenebrionidae	N		[195]
**Vip2Ad**	*Diabrotica virgifera*	Chrysomelidae	N		[191]
**Vip2Ae**	*Holotrichia oblita*	Scarabaeidae	N		[196]
	*Tenebrio molitor*	Tenebrionidae	N		[196]
**Vip2Ag**	*Anomala corpulenta*	Scarabaeidae	N		[192]
	*Holotrichia oblita*	Scarabaeidae	N		[192]
	*Holotrichia parallela*	Scarabaeidae	N		[192]
**Vip1Aa+Vip2Aa**	*Diabrotica longicornis B.*	Chrysomelidae	A		[185]
	*Diabrotica undecimpuntata*	Chrysomelidae	A		[185]
	*Diabrotica virgifera*	Chrysomelidae	A		[185]
	*Leptinotarsa decemlineata*	Chrysomelidae	N		[185]
	*Tenebrio molitor*	Tenebrionidae	N		[185]
**Vip1Aa+Vip2Ab**	*Diabrotica virgifera*	Chrysomelidae	A		[185]
**Vip1Ab+Vip2Aa**	*Diabrotica virgifera*	Chrysomelidae	N		[185]
**Vip1Ab+Vip2Ab**	*Diabrotica virgifera*	Chrysomelidae	N		[185]
**Vip1Ac+Vip2Ac**	*Tenebrio molitor*	Tenebrionidae	N		[195]
**Vip1Ac+Vip2Ae**	*Holotrichia oblita*	Scarabaeidae	N		[196]
	*Tenebrio molitor*	Tenebrionidae	N		[196]
**Vip1Ad+Vip2Ag**	*Anomala corpulenta*	Scarabaeidae	A	220 ng/g soil	[192]
	*Holotrichia oblita*	Scarabaeidae	A	120 ng/g soil	[192]
	*Holotrichia parallela*	Scarabaeidae	A	80 // 2.33 ng/g soil	[195,197]
**Vip1Ca+Vip2Aa**	*Tenebrio molitor*	Tenebrionidae	N		[187]
**Vip1Da+Vip2Ad**	*Diabrotica longicornis B.*	Chrysomelidae	A	213	[191]
	*Diabrotica undecimpuntata*	Chrysomelidae	A	4.91	[191]
	*Diabrotica virgifera*	Chrysomelidae	A	437	[191]
	*Leptinotarsa decemlineata*	Chrysomelidae	A	37	[191]
	*Anthonomus grandis*	Curculionidae	A	207	[191]
**Vip1Ba+Vip2Ba**	*Diabrotica virgifera*	Chrysomelidae	A		[190]
**Vip1Bb+Vip2Bb**	*Diabrotica virgifera*	Chrysomelidae	A		[190]
**Vip3Aa**	*Tenebrio molitor*	Tenebrionidae	N		[197]

^(a)^ The parameter is mortality. A = active; N = not active; ^(b)^ LC_50_ = lethal concentration that causes 50% mortality of the insects. data are expressed in μg/mL, unless otherwise stated. “//” separate two different values of the LC_50_.

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
