# Peer review of "Insecticidal Activity of *Bacillus thuringiensis* Proteins against Coleopteran Pests"

_toxins, 2020, doi:10.3390/toxins12070430_

Round 1
Reviewer 1 Report
Dear Authors, very good work you give some very good information about the Bt against coleoptera. The action of mode is very very good. My comments is in the pdf file.
Please check the refer numbering.

Author Response
Please see the attachment
Thank you very much for the comments

Reviewer 2 Report
General comments
Overall this review is provides a good overview of the literature on the insecticidal activity of Bt toxin against Coleoptera. However, as this more detail could be provided in sections 3 (Bt based insecticides), 4 (Bt-crops) and 5 (Resistance). All 3 sections are very brief.
The tables take up a disproportionate amount of text (approximately 5 pages out of 16 pages of text, excluding references). Could the tables be reduced in length?
For both tables 1 and 2 the LC50 values are provided for some, but not all, toxins that are active. It would be useful if some other measure of the degree of activity/potency could be provided for those examples with no LC50 value.
The supporting figures are good.
Specific comments
Abstract
Line 5. .. studied for many years due to its toxicity against the main insect pests…which are and of what?
Lines 55-56. Insert “which enable them to”: Beetles are the largest order in the class Insecta, and both, larvae and adults, have strong jaws, which enable them to feed on a wide variety of plant substrates,
Line 57. It is studies of Bt (toxins) against Coleoptera that are limited, not coleoptera themselves. This needs clarifying.
Section 2.1.1.
Three or 3, be consistent (three-domain or 3-domain)
A figure showing the conserved C-terminus could be included?
Section 2.3.1. Solubilization and Proteolytic Processing
Line 233. …to initiate the intoxication process. Is intoxication the correct word to sue here, normally associated with alcohol or drugs?
Section 3. Bt based insecticides
Lines 419-421.
However, this commercial product is useful for the control of other beetles, such as the chrysomelids Chrysophtharta bimaculata, C. agricola and C. scripta [202,203] or Lissorhoptrus oryzophilus (Coleoptera: Curculionidae) [204] under laboratory conditions.
Need to clarify which are field and which are laboratory studies. I would suggest that the product shouldn’t be described as useful if only tested in the laboratory.
Author Response
Please see the attachment
Thank you very much for the comments.
